# Matrix-Free GPU Semidefinite Programming for Quantum Ordered Search at the k=6 Frontier

Yancheng Wu[1]  Huikang Liu[2]  Wenzhi Gao[3]  Yuexin Su[4,5]  Tongyang Li[4,5]  Dongdong Ge[2,6]  Yinyu Ye[6,2,3]

## Abstract

Quantum computation offers the potential for a significant constant-factor speedup for the Ordered Search Problem (OSP). A classical construction is the $k$-query quantum ordered search algorithm, which can exactly search an $N$-element ordered list and achieves a query complexity improvement of a factor of $\frac{k}{\log_2 N}$. For larger $k$, stronger constant-factor improvements could be obtained by finding the largest admissible list size $N^\star$, a task that can be formulated as a structured semidefinite program (SDP). However, solving this SDP becomes computationally intractable beyond $k = 6$, as existing CPU and GPU solvers rely on explicit construction of prohibitively large constraint matrices. In this paper, we introduce a matrix-free GPU SDP framework that evaluates the highly structured constraints in OSP on-the-fly using custom CUDA kernels, reducing memory complexity from quadratic to linear and shifting the bottleneck from memory to computation. Using this approach, we provide strong numerical evidence that the optimal list size for $k = 6$ is at least $90,000$, improving the empirical upper bound on the query coefficient from $0.390$ to $0.365$. We further rigorously certify the upper bound $N^\star < 94,000$ by constructing dual infeasibility certificates via matrix-free minimum-eigenvalue estimation.

[1]School of Artificial Intelligence, Shanghai Jiao Tong University, Shanghai, China [2]Antai College of Economics and Management, Shanghai Jiao Tong University, Shanghai, China [3]Stanford University, USA [4]School of Computer Science, Peking University [5]Center on Frontiers of Computing Studies, Peking University [6]Shanghai Institute for Mathematics and Interdisciplinary Sciences, Shanghai, China. Correspondence to: Dongdong Ge <ddge@sjtu.edu.cn>, Huikang Liu <hkl1u@sjtu.edu.cn>.

*Proceedings of the 43rd International Conference on Machine Learning*, Seoul, South Korea. PMLR 306, 2026. Copyright 2026 by the author(s).

## 1. Introduction

The Ordered Search Problem (OSP) is a fundamental problem in computer science, concerned with locating a target element in a sorted list of $n$ items. Classical algorithms achieve optimal performance with a query complexity of $\log_2 n$. In contrast, quantum computation offers the possibility of a constant-factor speedup. Quantifying this advantage, typically expressed as a coefficient $c$ in the query complexity $c \log_2 n$, remains a central open problem in quantum algorithms.

Theoretical results have established tight asymptotic bounds on this coefficient. In particular, Hoyer & Neerbek (2001) showed that any exact quantum algorithm for OSP requires at least $\frac{\ln n - 1}{\pi} \approx 0.221 \log_2 n$ queries. On the other hand, constructive quantum algorithms seek to reduce the corresponding upper bound, with the goal of closing the gap between known lower and upper limits.

A seminal contribution by Childs et al. (2007) established a deep connection between quantum algorithm design and convex optimization. Specifically, they showed that the existence of an exact, $k$-query translation-invariant quantum algorithm for an ordered list of size $N$ is equivalent to the feasibility of a corresponding semidefinite program (SDP). This equivalence provides a constructive route for improving the upper bound on the query coefficient $c$: for a fixed number of queries $k$, identifying the largest integer $N^\star$ for which the SDP is feasible immediately yields the bound $c \leq k / \log_2 N^\star$. Using this framework, Childs et al. (2007) fully resolved the case $k = 4$, obtaining $N^\star = 605$.

The scaling behavior of this SDP, however, presents a major computational challenge. To mitigate the quadratic memory growth inherent in the original formulation, recent work has pursued alternative strategies. Carolan et al. (2025) proposed an innovative approach based on a linear programming (LP) relaxation, which enabled the problem to be solved for $k = 5$. This result established the current state of the art with $N^\star = 7265$, yielding the tightest known upper bound of $c \approx 0.390$.

A different line of work seeks to address the SDP directly using modern solvers that exploit low-rank factorizations of the matrix variables. GPU-accelerated methods such

as cuLoRADS (Han et al., 2024b) significantly reduce the effective dimensionality of the problem by representing semidefinite variables in factored form. This approach has proven highly effective for large-scale SDPs and offers a promising avenue for pushing the computational frontier of the OSP SDP beyond previously attainable regimes.

Indeed, for all known cases up to $k = 5$, a direct application of low-rank SDP solvers such as cuLoRADS is highly effective. However, advancing to the next frontier at $k = 6$, where the admissible list size $N$ is expected to be on the order of $10^5$, exposes a qualitatively different and more fundamental bottleneck. Although low-rank factorization keeps the number of optimization variables manageable, existing solvers still require explicit construction and storage of the SDP constraint matrix $\mathcal{A}$. At the scale of $k = 6$, materializing this matrix, even in sparse form, demands memory far beyond the capacity of modern high-performance GPUs, which leads to an insurmountable *memory wall*.

In this work, we overcome this bottleneck by developing a domain-aware, **matrix-free Augmented Lagrangian Method**. By implementing custom CUDA kernels that evaluate the highly structured constraints on-the-fly, we reduce the memory footprint of the constraints from prohibitive to negligible. This matrix-free design enables us to solve the OSP semidefinite program for $k = 6$ on a single GPU. Our main contributions are summarized as follows:

- **Matrix-Free GPU Optimization Framework.** We introduce a GPU-accelerated Augmented Lagrangian solver that entirely avoids explicit storage of the SDP constraint matrix. Through custom on-the-fly operator evaluation, the computational bottleneck is shifted from memory bandwidth to arithmetic throughput.

- **Resolution of the $k = 6$ Computational Frontier.** Using our framework, we bracket the optimal list size as $90,000 \leq N^\star < 94,000$. The lower bound is supported by high-precision numerical evidence, yielding a new upper bound on the quantum query complexity coefficient of $c \approx 0.365$. Moreover, the upper bound is mathematically proven: the infeasibility result at $N = 94,000$ is certified by a rigorous dual infeasibility certificate constructed via spectral shifting, ensuring mathematical validity despite the use of floating-point arithmetic.

SDP is a core computational primitive in many areas of scientific computing (see the related work), and the scalability and algorithmic generality of our approach make it well suited for extension to these applications.

## 1.1. Related Work

**SDP in Scientific Computing.** Semidefinite programming (SDP) is a central tool in quantum information science, where many physically meaningful optimization problems admit natural SDP formulations. A canonical example is quantum state discrimination, which seeks an optimal measurement strategy for identifying an unknown quantum state drawn from a known ensemble and can be formulated exactly as an SDP (Watrous, 2018; Skrzypczyk & Cavalcanti, 2023). SDP-based techniques also play an important role in quantum metrology and entanglement theory, for example in optimizing quantum Fisher information via iterative SDP schemes (Lukacs et al., 2025). In addition, certain quantum chemistry Hamiltonians admit sum-of-squares representations, enabling the characterization of spectral properties through SDP relaxations (Low et al., 2025).

**Large-Scale SDP Solvers.** Traditional SDP solvers, such as MOSEK (ApS, 2019) and COPT (Ge et al., 2022), are primarily based on interior-point methods and are highly effective for small- to medium-scale problems. However, their reliance on sequential, memory-intensive matrix factorizations fundamentally limits their scalability, particularly on modern GPU architectures.

To overcome these limitations, GPU-accelerated first-order methods based on the Burer–Monteiro low-rank factorization framework (Burer & Monteiro, 2003) have emerged as a powerful alternative for large-scale SDPs. Representative solvers include LoRADS (Han et al., 2024a;b), ALORA (Ding et al., 2025), and cuHALLaR (Aguirre et al., 2025), which replace expensive factorizations with low-rank updates and exploit massive GPU parallelism. These methods have demonstrated the ability to solve classical SDP benchmarks, such as Max-Cut, at unprecedented scales.

More broadly, GPU-accelerated primal–dual first-order methods have achieved notable success as general-purpose solvers for large-scale convex optimization. Examples include PDLP and HPR-LP for linear programming (Applegate et al., 2021; Lu & Yang, 2023a; Lu et al., 2025; Li et al., 2026; Chen et al., 2024), rAPDHG, PDHCG, and HPR-QP for quadratic programming (Lu & Yang, 2023b; Huang et al., 2024; Chen et al., 2025), PDCS for general conic optimization (Lin et al., 2026), PDOT and HPR-OT for optimal transport (Lu & Yang, 2024; Zhang et al., 2025), as well as large-scale market equilibrium and network flow problems (Liu et al., 2025; Zhang & Boyd, 2025). By combining simple iterative updates with operator splitting and GPU parallelism, these methods substantially outperform traditional solvers in scalability and efficiency on large-scale instances.

## 2. Problem Formulation and Preliminaries

### 2.1. Quantum Ordered Search

In the ordered search problem, given an ordered sequence $\{a_1, a_2, \cdots, a_n\}$, one is given oracle access to a monotone Boolean function

$$f(i) = \mathbf{1}\{x \le a_i\}, \qquad i \in 1, \ldots, n,$$

and the goal is to identify the unique transition point. Classical decision-tree arguments imply that each comparison query yields at most one bit of information, leading to a worst-case lower bound of $\log_2 n$ queries.

In the quantum query model, a single oracle call

$$O_f : |i\rangle \mapsto (-1)^{f(i)}|i\rangle$$

acts coherently on a superposition of indices. By combining such phase queries with carefully designed unitary transformations, a quantum algorithm can use interference to eliminate a larger fraction of candidate indices per query than is possible classically. As a result, the quantum query complexity satisfies

$$Q(n) = c \log_2 n + O(1), \qquad c < 1,$$

with known lower bounds $c \gtrsim 0.22$ (Hoyer & Neerbek, 2001) and explicit algorithms achieving $c \approx 0.39$ (Carolan et al., 2025). Thus, quantum ordered search achieves a provable constant-factor speedup over classical binary search, while respecting the information-theoretic $\Theta(\log n)$ lower bound imposed by the problem structure.

### 2.2. The $k$-Query Algorithm and Its SDP Formulation

Most known upper bounds are derived from the recursive $k$-query algorithm, which exactly searches an ordered list of size $N$ and yields a query complexity of

$$k \log_N n = \frac{k}{\log_2 N} \log_2 n.$$

Childs et al. (2007) reformulated this approach using symmetric Laurent polynomials to track the evolution of quantum amplitudes across queries. By invoking the Fejér–Riesz theorem, the algorithm's correctness and unitarity conditions are converted into positivity constraints on Toeplitz matrices, leading to an exact semidefinite programming (SDP) formulation whose feasible solutions correspond precisely to valid $k$-query algorithms.

In particular, the existence of an exact $k$-query quantum algorithm for an ordered list of size $N$ is equivalent to the feasibility of finding a sequence of positive semidefinite matrices $Q_0, \ldots, Q_k \in \mathbb{S}^N$ satisfying

$$Q_0 = E/N, \qquad Q_k = I/N,$$
$$\mathrm{Tr}(Q_t) = 1, \qquad \forall\, t \in \{1, \ldots, k-1\}, \qquad (1)$$
$$\mathcal{T}_t(Q_t) = \mathcal{T}_t(Q_{t-1}), \qquad \forall\, t \in \{1, \ldots, k\}. \qquad (2)$$

Here, $E$ denotes the all-ones matrix and $I$ the identity matrix. The linear evolution operator $\mathcal{T}_t : \mathbb{S}^N \to \mathbb{R}^{N-1}$ is defined component-wise: for $i \in \{1, \ldots, N-1\}$,

$$(\mathcal{T}_t X)_i := \langle T_{t,i}, X \rangle = \mathrm{Tr}_i(X) + (-1)^t \mathrm{Tr}_{i-N}(X), \quad (3)$$

where $T_{t,i} \in \mathbb{R}^{N \times N}$ is the atomic matrix that extracts the corresponding diagonal entries. Specifically, $T_{t,i}$ has ones on the $i$-th super-diagonal and $(-1)^t$ on the $(N - i)$-th sub-diagonal. The diagonal trace operator $\mathrm{Tr}_j(\cdot)$ is defined as

$$\mathrm{Tr}_j(X) = \begin{cases} \sum_{\ell=1}^{N-j} X_{\ell, \ell+j}, & j \ge 0, \\ \sum_{\ell=1}^{N+j} X_{\ell-j, \ell}, & j < 0. \end{cases} \qquad (4)$$

To apply general-purpose solvers, we condense the variable matrices $Q_1, \ldots, Q_{k-1}$ into a single block-diagonal decision variable $\mathbf{X} = \mathrm{diag}(Q_1, \ldots, Q_{k-1}) \in \mathbb{S}^{(k-1)N}$. The problem is then cast into the standard SDP form:

$$\text{find } \mathbf{X} \in \mathbb{S}^{(k-1)N} \quad \text{s.t.} \quad \mathcal{A}(\mathbf{X}) = b, \quad \mathbf{X} \succeq 0. \quad (5)$$

where the linear operator $\mathcal{A}$ is defined by a set of $m$ constraint matrices $\{A_1, \ldots, A_m\} \subset \mathbb{S}^{(k-1)N}$.

The total number of scalar constraints is $m = k(N - 1) + (k - 1)$. The constraint matrices $A_j$ and the right-hand side vector $b$ are constructed explicitly from the atomic matrices $T_{t,i}$ and the identity $I$, inheriting a highly sparse and block-diagonal structure. We provide the detailed explicit construction of dimensions, the vector $b$, and the matrices $\{A_j\}$ in Appendix A.

### 2.3. Low-Rank Factorization Strategy

The SDP arising from OSP exhibits a crucial structural property: whenever a feasible solution exists, there also exists a rank-one feasible solution. This allows the search space to be restricted without loss of generality. The result, originally established in the context of nonnegative Laurent polynomials (Dumitrescu, 2007; Carolan et al., 2025), is summarized below.

**Theorem 2.1** (Existence of Rank-1 Solution (Dumitrescu, 2007; Carolan et al., 2025))**.** *Let*

$$\mathcal{Q} = \left\{ Q \in \mathbb{S}_+^N \mid \mathrm{Tr}_j(Q) = a_j, \ j = -(N-1), \ldots, N-1 \right\}.$$

*If $\mathcal{Q}$ is non-empty, then there exists some $Q^\star \in \mathcal{Q}$ with* $\mathrm{rank}(Q^\star) = 1$.

Motivated by this result, we adopt the Burer–Monteiro factorization $Q_t = R_t R_t^\top$ with $R_t \in \mathbb{R}^{N \times r}$. Although Theorem 2.1 guarantees that $r = 1$ is sufficient in principle, the resulting nonconvex problem may exhibit poor local geometry. In practice, we therefore employ a small rank $r \ge 1$ to improve optimization robustness.

This reformulation converts the original convex SDP into a nonconvex problem with $O(Nr)$ variables, compared to $O(N^2)$ in the original formulation. While this low-rank representation resolves the memory bottleneck associated with the decision variables, it does not eliminate the dominant cost of storing the constraint matrix $\mathcal{A}$. For OSP at $k = 6$, $\mathcal{A}$ remains prohibitively large even under low-rank factorization. In Section 3, we introduce a matrix-free approach that removes this remaining bottleneck.

## 3. Matrix-Free Optimization on GPUs

To overcome the prohibitive memory cost encountered at the $k = 6$ frontier, we develop a *matrix-free* Augmented Lagrangian Method (ALM). Unlike standard SDP solvers that explicitly materialize the constraint matrix $\mathcal{A}$, our approach exploits the algebraic structure of the OSP constraints to evaluate the action of the associated linear operators implicitly. This section describes the resulting optimization framework and the decomposition of the key operators.

### 3.1. Matrix-Free ALM Framework

We consider the low-rank SDP formulation introduced in Section 2.3. The optimization variable is the factor matrix $R \in \mathbb{R}^{(k-1)N \times r}$, and the problem is formulated as the minimization of the Augmented Lagrangian

$$\mathcal{L}_\rho(R, \lambda) = \langle \lambda, \mathcal{A}(\mathbf{X}) - b \rangle + \frac{\rho}{2} \| \mathcal{A}(\mathbf{X}) - b \|^2, \quad (6)$$

where $\mathbf{X}$ is the block-diagonal matrix with blocks $R_t R_t^\top$, and $\lambda = (y, z)$ denotes the dual variables. Since the OSP SDP is a feasibility problem, the primal objective term vanishes.

The ALM alternates between primal and dual updates:

**Primal update:** $R^{k+1} = \arg\min_{R} \mathcal{L}_\rho(R, \lambda^k), \quad (7)$

**Dual update:** $\lambda^{k+1} = \lambda^k + \rho(\mathcal{A}(\mathbf{X}^{k+1}) - b). \quad (8)$

The dominant computational cost arises in the primal update (7), which solves an unconstrained nonconvex subproblem using L-BFGS. To avoid the $O(N^2)$ memory cost of explicitly forming $\mathcal{A}$, we identify two linear operators that must be evaluated efficiently in a matrix-free manner.

**1. Forward Operator $\mathcal{A}(\cdot)$.** Evaluating the Augmented Lagrangian during line search requires mapping low-rank matrices to the constraint space. For a generic product $UV^\top$, the operator decomposes as follows:

- **Trace constraints:** $\mathrm{Tr}(UV^\top) = \langle U, V \rangle$, which is a simple inner product.

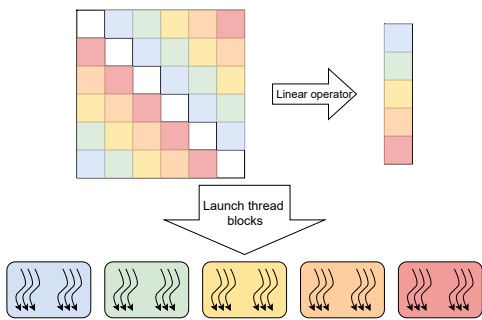

*Figure 1.* **Structure-Aware Parallel Mapping.** *Top:* The operator $\mathcal{T}_t$ aggregates matrix entries along specific diagonals (color-coded) to form $v$. *Bottom:* Each component $v_j$ is computed by an independent parallel unit, avoiding explicit matrix materialization.

- **Evolution constraints:** $\mathcal{T}_t(UV^\top)$, which involves summing selected diagonals of a dense matrix product and constitutes the main computational bottleneck.

In Section 3.2, we show how to evaluate these diagonal sums implicitly by mapping them directly to parallel compute units, eliminating the need to materialize intermediate matrices.

**2. Adjoint Operator $\mathcal{A}^*(\cdot)R$.** Computing the gradient of the Augmented Lagrangian requires applying the adjoint operator:

$$\nabla_R \mathcal{L}_\rho = 2\,\mathcal{A}^*\Big(\lambda + \rho\big(\mathcal{A}(\mathbf{X}) - b\big)\Big)R. \quad (9)$$

For each block $R_t$, the adjoint action takes the structured form

$$\big(\mathcal{A}^*(y, z)R\big)_t = z_t R_t + \big((\mathcal{T}_t^*(y_t))^{\mathrm{sym}} - (\mathcal{T}_{t+1}^*(y_{t+1}))^{\mathrm{sym}}\big)R_t,$$

where $y, z$ are corresponding to evolution constraints (2) and trace constraints (1), $\mathcal{T}_t^*(y_t) = \sum_{i=1}^{N-1} y_i T_{t,i}$ and $(\mathcal{T}_t^*(y_t))^{\mathrm{sym}} = \frac{1}{2}(\mathcal{T}_t^*(y_t) + (\mathcal{T}_t^*(y_t))^T)$. While the scalar scaling is trivial, explicitly forming the dense matrix $\mathcal{T}_t^*(y)$ is infeasible at scale. In Section 3.3, we exploit its near-circulant structure to implement this operation as a memory-efficient, convolution-like kernel, completing the matrix-free ALM design.

### 3.2. Implicit Evaluation of the Forward Operator

Evaluating the evolution constraints $\mathcal{T}_t(X)$ by explicitly forming $X = UV^\top$ incurs an $O(N^2)$ memory cost and is infeasible at scale. To avoid this bottleneck, we define an equivalent implicit operator

$$\hat{\mathcal{T}}_t : \mathbb{R}^{N \times r} \times \mathbb{R}^{N \times r} \to \mathbb{R}^{N-1},$$

which acts directly on the low-rank factors $U$ and $V$. Substituting the low-rank expansion $X_{ij} = \sum_{d=1}^{r} U_{i,d} V_{j,d}$ yields

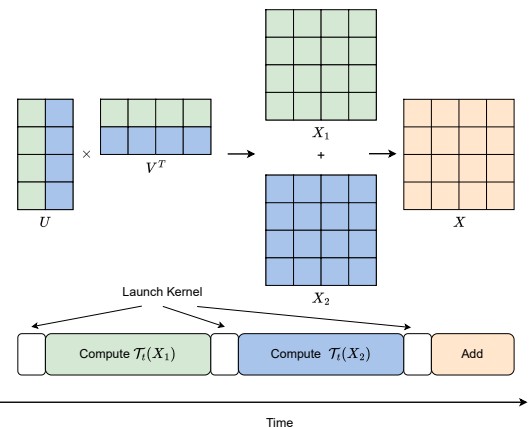

*Figure 2.* **Memory Optimization.** By decomposing computation into rank-1 updates $\mathcal{T}(UV^\top) = \sum_d \mathcal{T}(u_d v_d^\top)$, we reduce the working set from $O(Nr)$ to $O(N)$, enabling execution in on-chip memory.

the $j$-th component of $v = \hat{\mathcal{T}}_t(U, V)$:

$$v_j = \sum_{l=1}^{N-j} \sum_{d=1}^{r} U_{l,d} V_{l+j,d} + (-1)^t \sum_{l=1}^{j} \sum_{d=1}^{r} U_{l+N-j,d} V_{l,d}, \tag{10}$$

for $j = 1, \ldots, N-1$. Each $v_j$ depends only on a single diagonal of $X$ and can therefore be computed independently. This property enables a natural parallelization strategy: each component is assigned to a separate compute unit, with the inner sums evaluated via parallel reduction (see Figure 1), allowing fully on-the-fly computation without matrix materialization.

**Memory Optimization via Linearity.** While (10) avoids storing $X$, a naive implementation still accesses all rank-$r$ factors, resulting in an $O(Nr)$ working set. To reduce memory traffic, we exploit the linearity of $\hat{\mathcal{T}}_t$ and swap the order of summation:

$$v_j = \sum_{d=1}^{r} \left( \sum_{l=1}^{N-j} U_{l,d} V_{l+j,d} + (-1)^t \sum_{l=1}^{j} U_{l+N-j,d} V_{l,d} \right). \tag{11}$$

This reformulation expresses $\hat{\mathcal{T}}_t$ as a sum of rank-1 contributions. Each term accesses only the $d$-th columns of $U$ and $V$, reducing the active working set to $O(N)$. As illustrated in Figure 2, this column-wise streaming fits entirely in fast on-chip memory, substantially increasing arithmetic intensity and alleviating memory bandwidth bottlenecks.

### 3.3. Implicit Evaluation of the Adjoint Operator

As shown in Section 3.1, gradient evaluation requires computing $L = \mathcal{T}_t^*(y)R$, where $y \in \mathbb{R}^{N-1}$ and $R \in \mathbb{R}^{N \times r}$. Explicitly forming the dense matrix $\mathcal{T}_t^*(y) \in \mathbb{R}^{N \times N}$ incurs

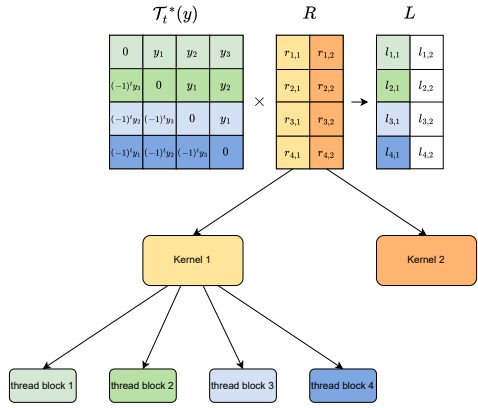

*Figure 3.* **Parallel Evaluation.** The product $\mathcal{T}_t^*(y)R$ is computed column-by-column via modified convolution. A dedicated unit handles the cyclic shifts and parity-dependent sign flip on-the-fly, avoiding matrix construction.

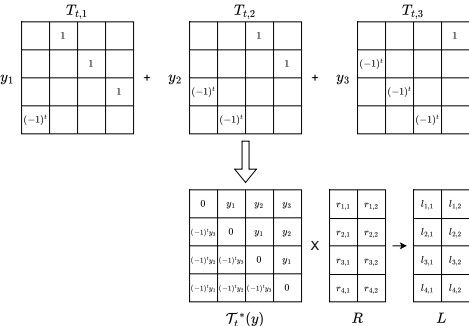

*Figure 4.* **Structural Synthesis.** The dense matrix $\mathcal{T}_t^*(y)$ is a weighted sum of atomic matrices, forming a near-circulant structure defined by $y$ and parity $(-1)^t$. This allows for a fully implicit representation.

an $O(N^2)$ memory cost and is infeasible at scale. We therefore evaluate the adjoint action in a matrix-free manner by exploiting its algebraic structure.

The adjoint operator can be written as $\mathcal{T}_t^*(y) = \sum_{i=1}^{N-1} y_i T_{t,i}$, where each atomic matrix $T_{t,i}$ has ones on the $i$-th super-diagonal and $(-1)^t$ on the $(N-i)$-th sub-diagonal. As a result, $\mathcal{T}_t^*(y)$ exhibits a parity-dependent Toeplitz (near-circulant) structure:

$$(\mathcal{T}_t^*(y))_{k,j} = \begin{cases} y_{j-k}, & j > k, \\ (-1)^t y_{N+j-k}, & j < k. \end{cases} \tag{12}$$

Hence, the entire operator is fully characterized by the vector $y$ and the parity of $t$, requiring only $O(N)$ storage (Figure 4).

**Modified Convolution.** Using (12), the product $L = \mathcal{T}_t^*(y)R$ reduces to a sequence of column-wise modified convolutions. For a column $v$ of $R$, the output $l = \mathcal{T}_t^*(y)v$ satisfies

$$l_k = \sum_{j=k+1}^{N} y_{j-k}v_j + (-1)^t \sum_{j=1}^{k-1} y_{N+j-k}v_j. \qquad (13)$$

Each entry $l_k$ is independent and can be computed by a dedicated parallel unit via reduction (Figure 3). By streaming the columns of $R$, the adjoint action is evaluated on-the-fly with linear memory complexity and high arithmetic intensity, without explicitly forming $\mathcal{T}_t^*(y)$.

# 4. Infeasibility Certification

While the ALM solver provides strong empirical evidence of infeasibility through stagnation of the primal residuals, a rigorous conclusion requires a valid dual certificate.

### 4.1. Dual Formulation and Certification

To establish the non-existence of a quantum algorithm, we consider the dual SDP. The dual variables consist of vectors $y_t \in \mathbb{R}^{N-1}$ associated with the evolution constraints and scalars $z_t \in \mathbb{R}$ associated with the trace constraints. The dual problem is given by

$$\begin{aligned} \max \quad & \mathrm{Obj}(y, z) = \langle y_1, \mathcal{T}_1(Q_0) \rangle + \sum_{t=1}^{k-1} z_t \\ \text{s.t.} \quad & S_t = -\Big( (\mathcal{T}_t^*(y_t))^{\mathrm{sym}} - (\mathcal{T}_{t+1}^*(y_{t+1}))^{\mathrm{sym}} + z_t I \Big), \\ & S_t \succeq 0, \qquad t = 1, \ldots, k-1, \end{aligned}$$
$$(14)$$

where $\mathcal{T}_t^*$ denotes the adjoint evolution operator and $(\cdot)^{\mathrm{sym}}$ extracts the symmetric part. Each slack matrix $S_t \in \mathbb{S}^N$ admits a structured Toeplitz-like form (see Appendix C).

By the theorem of strong alternatives, the existence of a dual feasible point $(y, z)$ with $\mathrm{Obj}(y, z) > 0$ certifies strict infeasibility of the primal SDP.

### 4.2. Spectral Repair Strategy

Numerical solvers operate in finite precision and typically produce approximate dual solutions $(\tilde{y}, \tilde{z})$ with $\mathrm{Obj}(\tilde{y}, \tilde{z}) > 0$, while the corresponding slack matrices $S_t$ may exhibit small negative eigenvalues, invalidating feasibility. To address this issue, we apply a *spectral repair strategy* that shifts the spectrum of each $S_t$ upward to restore positive semidefiniteness at the cost of reducing the dual objective.

**Theorem 4.1** (Sufficient Condition for Infeasibility)**.** *The primal SDP is strictly infeasible if there exists $(\tilde{y}, \tilde{z})$ such*

*that*

$$\mathrm{Obj}(\tilde{y}, \tilde{z}) > \sum_{t=1}^{k-1} \max\big(0, -\lambda_{\min}(S_t)\big). \qquad (15)$$

The proof, given in Appendix C, shows that this condition guarantees the existence of a repaired dual solution that is strictly feasible with positive objective value.

### 4.3. Matrix-Free Verification

The main computational challenge is evaluating $\lambda_{\min}(S_t)$ for $N \approx 10^5$, since $S_t$ is dense. Exploiting the Toeplitz-like structure detailed in Appendix C, we employ a **matrix-free Lanczos method**. The required matrix–vector product $w = S_t v$ is implemented as a convolution-like operation using custom CUDA kernels. This enables efficient computation of the minimum eigenvalues and repair costs on a single GPU, without ever materializing the dense slack matrices.

# 5. Experiments and Results

We evaluate our matrix-free approach in two stages: (i) validating the integrated solver on known instances with $k \leq 5$, and (ii) solving the large-scale $k = 6$ problem to demonstrate that the memory barrier can be overcome.

### 5.1. Experimental Setup and Kernel Performance

All experiments were conducted on a single NVIDIA H100 GPU (80GB VRAM) using CUDA 12.6. Our implementation extends the cuLoRADS framework in Julia and integrates custom CUDA C++ kernels to support matrix-free operator evaluation.

We first benchmark the forward and adjoint kernels on a large-scale instance ($k = 6$, $N = 100,000$, $r = 1$). As reported in Table 1, both operators complete in approximately 71 ms and require only about 560 MB of GPU memory. In contrast, explicitly storing the corresponding constraint matrices would require multiple terabytes of VRAM. These results confirm that our approach successfully eliminates the memory wall.

*Table 1.* Performance of custom CUDA kernels for $k = 6$, $N = 100,000$, $r = 1$. Both operators achieve millisecond-scale latency with negligible memory overhead.

| Operation | Time (ms) | Memory (MB) |
|---|---|---|
| Forward operator $\mathcal{A}(RR^\top)$ | 71.02 | 551.06 |
| Adjoint product $\mathcal{A}^*(y)R$ | 71.18 | 561.06 |

### 5.2. Integrated Matrix-Free Solver Performance

We next replace the standard matrix-based operators in the cuLoRADS ALM module with our custom kernels, yielding a fully matrix-free solver.

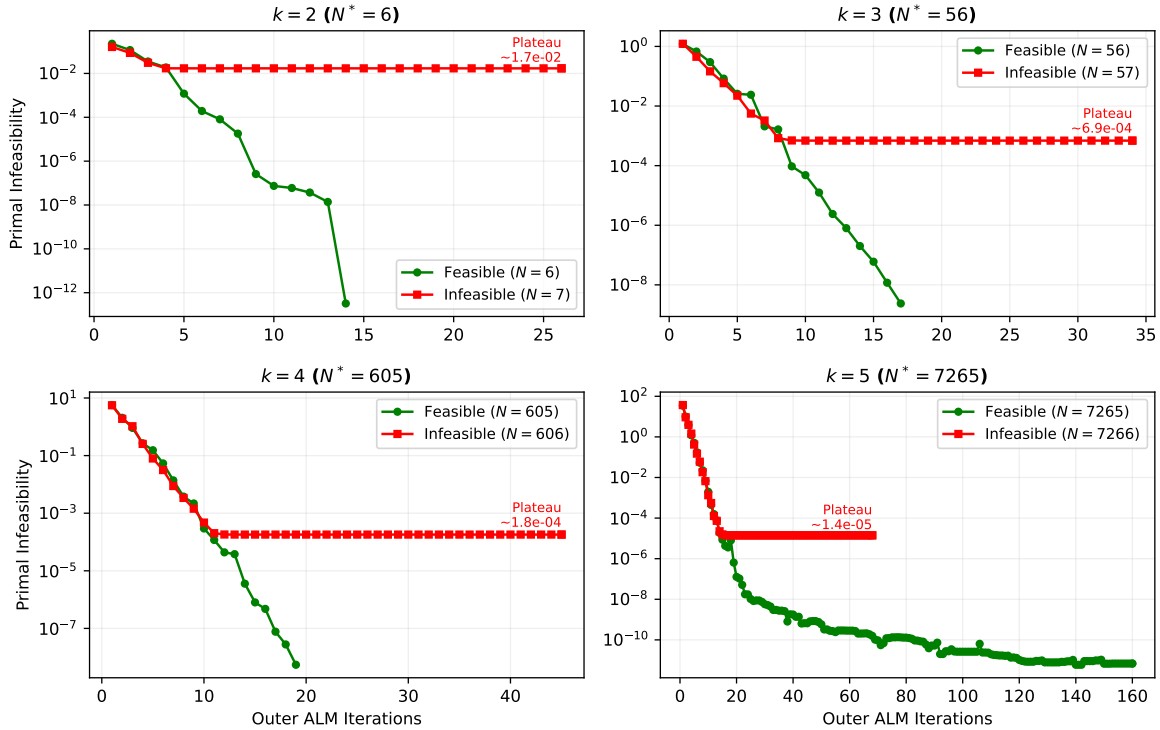

*Figure 5.* **Empirical Phase Transition** ($k \leq 5$). The green curves ($N^\star$) show rapid convergence to high precision, while the red curves ($N^\star + 1$) consistently hit a numerical "infeasibility floor." This uniform behavior validates the solver's robustness across problem scales.

### 5.2.1. VALIDATION ON KNOWN CASES ($k \leq 5$)

We first validate correctness by reproducing the known optimal list sizes $N^\star$ for $k \in \{2, 3, 4, 5\}$. For each $k$, we test two instances: the optimal size $N^\star$ and the immediately infeasible instance $N^\star + 1$.

Figure 5 shows a clear phase transition. For all feasible instances (green curves), the primal infeasibility decays to numerical zero. In contrast, all infeasible instances (red curves) exhibit stagnation at a nonzero *infeasibility floor*, as expected for Augmented Lagrangian methods. The magnitude of this floor decreases with $k$, ranging from $\approx 10^{-2}$ for $k = 2$ to $\approx 1.4 \times 10^{-5}$ for $k = 5$. These results confirm that the matrix-free formulation preserves the original optimization landscape and reliably distinguishes feasibility from infeasibility.

### 5.2.2. BREAKING THE BARRIER OF $k = 6$ INSTANCE

We finally apply our solver to the previously intractable $k = 6$ case. All experiments were performed on a single H100 GPU, with a total runtime of approximately 16 days.

**Feasibility at $N = 90,000$.** Table 2 summarizes the convergence behavior for the critical instance $N = 90,000$. The solver reduces the relative primal infeasibility to $7.49 \times 10^{-10}$ and absolute primal infeasibility to $1.99 \times 10^{-10}$. We consider this high-precision convergence and the distinct absence of the "infeasibility plateau" as highly compelling numerical evidence for exact feasibility, yielding a new upper bound on the quantum query complexity coefficient:

$$c \leq 6/\log_2(90,000) \approx 0.365. \tag{16}$$

*Table 2.* Selected convergence log for $k = 6$, $N = 90,000$. Both the relative primal infeasibility and the maximum constraint violation decrease to $\approx 10^{-10}$, confirming feasibility.

| Iter | Rel. Infeasibility $\|\mathcal{A}(\mathbf{X}) - b\|_2/\|b\|_\infty$ | Max. Violation $\|\mathcal{A}(\mathbf{X}) - b\|_\infty$ | Time (s) |
|------|------|------|------|
| 1 | 1.49e+02 | 1.18e+02 | 1.3e1 |
| 20 | 1.34e-05 | 5.03e-06 | 3.7e5 |
| 50 | 1.59e-08 | 5.65e-09 | 1.2e6 |
| 94 | 7.49e-10 | 1.99e-10 | 1.4e6 |

**Infeasibility at $N = 94,000$.** To bracket the optimal list size, we next consider the instance $N = 94,000$. Figure 6 overlays the primal infeasibility trajectories for both

cases. The $N = 94,000$ run stagnates at a clear infeasibility plateau of approximately $2.2 \times 10^{-3}$. This behavior closely mirrors the infeasible regimes observed for smaller values of $k$ and provides strong numerical evidence that $N = 94,000$ is infeasible.

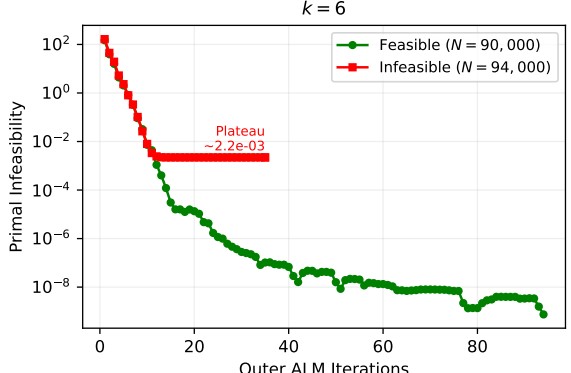

*Figure 6.* **Phase Transition at** $k = 6$**.** The feasible case ($N = 90,000$, green) converges to zero primal infeasibility, whereas the $N = 94,000$ case (red) stagnates at a nonzero plateau, indicating infeasibility.

**Rigorous Certification.** To convert this numerical evidence into a formal proof, we apply the spectral shifting strategy introduced in Section 4. At each iteration, we normalize the dual variables $(\hat{y}, \hat{z})$ and compute the certification gap

$$\text{Gap} = \underbrace{\text{Obj}(\hat{y}, \hat{z})}_{\text{dual objective}} - \underbrace{\sum_{t=1}^{k-1} \max(0, -\lambda_{\min}(S_t))}_{\text{repair cost}}. \quad (17)$$

Figure 7 illustrates the evolution of the dual objective and the spectral repair cost. Initially, the repair cost exceeds the dual objective, yielding a negative gap. As the solver progresses, the repair cost decreases rapidly. Starting from iteration 18, the gap becomes strictly positive, indicating that the dual objective is sufficient to absorb the repair cost. This establishes the existence of a valid dual infeasibility certificate and rigorously proves that no exact 6-query quantum algorithm exists for $N = 94,000$.

Combining the feasibility result at $N = 90,000$ with the certified infeasibility at $N = 94,000$, we conclude that the optimal list size for $k = 6$ satisfies

$$90,000 \leq N^\star < 94,000.$$

**Conjecture.** We further examine two intermediate values, $N = 92,000$ and $N = 93,000$, whose primal infeasibility trajectories are reported in Figure 8. For $N = 92,000$,

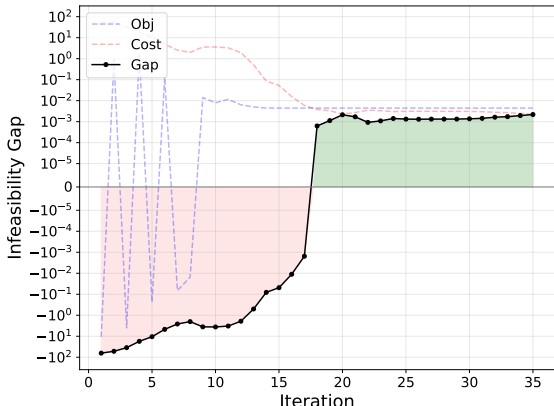

*Figure 7.* **Rigorous Dual Certification for** $N = 94,000$**.** The normalized dual objective (**Obj**) and the spectral repair cost (**Cost**) are tracked across iterations. The infeasibility **Gap** (Obj − Cost) becomes strictly positive at iteration 18 (green shaded region), certifying primal infeasibility.

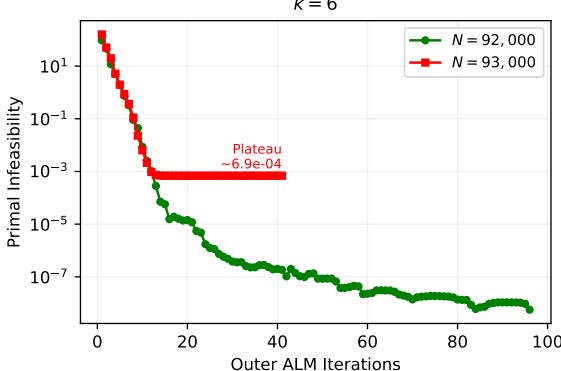

*Figure 8.* **Empirical investigation of conjecture.** $N = 92,000$ shows feasible behavior, while $N = 93,000$ exhibits an infeasible plateau.

the primal infeasibility decreases steadily into the $10^{-8}$ regime. Although this residual is slightly larger than that observed in the verified feasible case $N = 90,000$, the absence of a stagnation plateau suggests that the instance is likely feasible, potentially requiring additional iterations or higher numerical precision for full resolution.

In contrast, the run with $N = 93,000$ exhibits clear stagnation at approximately $6.9 \times 10^{-4}$, consistent with the characteristic *infeasibility floor* observed in confirmed infeasible instances. However, unlike the case $N = 94,000$, our spectral shifting procedure was unable to construct a valid dual certificate for $N = 93,000$: the cumulative repair cost $\sum_t \epsilon_t$ remained slightly larger than the corresponding dual objective value. Based on these distinct patterns, we conjecture that the true maximum list size satisfies

$$92,000 \leq N^\star < 93,000.$$

## 6. Conclusion

We overcome the memory bottleneck at the $k = 6$ frontier of Quantum Ordered Search by developing a matrix-free Augmented Lagrangian solver that reduces constraint memory from quadratic to linear. This enables solving instances with $N \approx 10^5$ on a single GPU, bracketing the optimal list size as $90,000 \leq N^\star < 94,000$ and tightening the quantum query complexity coefficient to $c \approx 0.365$, with infeasibility certified via spectral shifting.

Beyond Quantum Ordered Search, our approach demonstrates the potential of *operator-based* convex optimization for scaling large, structured SDPs. Extending this framework to multi-GPU environments and further refining the bound on $N^\star$ remain important directions for future work.

## Acknowledgements

This research is partially supported by the National Natural Science Foundation of China (NSFC) under Grant No. 72225009, 72394360, 72394365, 12301403, 72192830, and 72192832.

## Impact Statement

This paper presents work whose goal is to advance the field of machine learning. There are many potential societal consequences of our work, none of which we feel must be specifically highlighted here.

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

## A. Explicit Construction of SDP Constraints

In this appendix, we detail the dimensions and explicit matrix structures for the SDP formulation presented in Section 2.2.

**Constraint Dimensions and Structure.** The system dimensions are precise. The total number of scalar linear constraints is $m = k(N-1) + (k-1)$. This sum comprises $k$ sets of evolution constraints (2), each of size $N-1$, and $k-1$ trace constraints (1) acting on the variable matrices.

The right-hand side vector $b \in \mathbb{R}^m$ is highly sparse and structured:

$$b = [\underbrace{(\mathcal{T}_1 Q_0)^T}_{N-1}, \underbrace{\mathbf{0}, \ldots, \mathbf{0}}_{(k-1)(N-1)}, \underbrace{1, \ldots, 1}_{k-1}]^T. \tag{18}$$

The first block arises from the boundary condition $\mathcal{T}_1 Q_1 = \mathcal{T}_1 Q_0$. The middle blocks are zero because the intermediate evolution constraints are homogeneous ($\mathcal{T}_t Q_t - \mathcal{T}_t Q_{t-1} = 0$) and the final boundary condition vanishes ($\mathcal{T}_k Q_k = \mathcal{T}_k(I/N) = \mathbf{0}$). The final ones enforce the unit trace.

**Explicit Construction of Constraint Matrices.** Crucially, each $A_j$ is a block-diagonal matrix constructed from the atomic matrices $T_{t,i}$ and the identity $I$. They fall into these categories:

1. *Initial Evolution ($t = 1$):* Constraints on $Q_1$. The matrices have the form $A_j = \text{diag}(T_{1,i}^{\text{sym}}, 0, \ldots, 0)$.

2. *Intermediate Evolution ($1 < t < k$):* Constraints linking $Q_{t-1}$ and $Q_t$. The matrices have non-zeros in two blocks:

$$A_j = \text{diag}(0, \ldots, \underbrace{-T_{t,i}^{\text{sym}}}_{\text{block } t-1}, \underbrace{T_{t,i}^{\text{sym}}}_{\text{block } t}, \ldots, 0). \tag{19}$$

3. *Final Evolution ($t = k$):* Constraints on $Q_{k-1}$. The matrices have the form $A_j = \text{diag}(0, \ldots, 0, -T_{k,i}^{\text{sym}})$.

4. *Trace Constraints:* $A_j = \text{diag}(0, \ldots, I, \ldots, 0)$, enforcing $\text{Tr}(Q_t) = 1$.

Since $T_{t,i}$ is not symmetric, we need to construct the symmetric constraint matrix with $T_{t,i}^{\text{sym}} = \frac{1}{2}(T_{t,i} + T_{t,i}^T)$.

## B. Proof of Rank-1 Solution Existence

In this appendix, we provide the proof for Theorem 2.1, demonstrating that the generalized trace constraints allow for a rank-1 solution without loss of generality. This result justifies the use of the low-rank factorization strategy employed in the main text.

**Theorem B.1** (Restated). *Let $\mathcal{Q}$ be the set of feasible solutions defined by fixing the generalized traces to specific values $\{a_j\}$:*

$$\mathcal{Q} = \left\{ Q \in \mathbf{S}_+^N \mid \text{tr}_j(Q) = a_j, \quad \forall j \in \{-(N-1), \ldots, N-1\} \right\}. \tag{20}$$

*If $\mathcal{Q}$ is non-empty, then there exists a matrix $Q^* \in \mathcal{Q}$ such that $\text{rank}(Q^*) = 1$.*

*Proof.* Consider the following optimization problem over the feasible set $\mathcal{Q}$:

$$\text{maximize} \quad Q_{0,0} \quad \text{subject to} \quad Q \in \mathcal{Q}. \tag{21}$$

The set $\mathcal{Q}$ is the intersection of the cone of positive semidefinite matrices (a closed set) with a set of affine hyperplanes defined by the trace constraints. Furthermore, since $\text{tr}_0(Q) = \text{tr}(Q) = a_0$ is fixed and $Q \succeq 0$, the Frobenius norm of $Q$ is bounded. Thus, $\mathcal{Q}$ is compact. The continuous function $f(Q) = Q_{0,0}$ therefore attains its maximum on $\mathcal{Q}$. Let $Q^*$ be a maximizer of this problem.

We proceed by contradiction. Assume that $\text{rank}(Q^*) > 1$.

**1. Block Decomposition and Schur Complement**    We partition $Q^*$ as follows:

$$Q^* = \begin{pmatrix} \alpha & v^\top \\ v & \hat{Q} \end{pmatrix}, \tag{22}$$

where $\alpha = Q^*_{0,0}$ is the scalar we are maximizing, $v \in \mathbb{R}^{N-1}$, and $\hat{Q} \in \mathbf{S}^{N-1}_+$. Since $Q^* \succeq 0$, by the properties of the Schur complement, the matrix $P$ defined below must be positive semidefinite:

$$P := \hat{Q} - \frac{1}{\alpha} v v^\top \succeq 0. \tag{23}$$

Using this definition, we can decompose $Q^*$ into a rank-1 component and a residual component:

$$Q^* = \underbrace{\begin{pmatrix} \alpha & v^\top \\ v & \frac{1}{\alpha} v v^\top \end{pmatrix}}_{\text{Rank-1}} + \underbrace{\begin{pmatrix} 0 & 0 \\ 0 & P \end{pmatrix}}_{\text{Residual}}. \tag{24}$$

From the rank additivity property of the Schur complement, $\operatorname{rank}(Q^*) = 1 + \operatorname{rank}(P)$. Since we assumed $\operatorname{rank}(Q^*) > 1$, it follows that $P \neq 0$.

**2. Constructing a Better Solution**    We now construct a new matrix $Q'$ by moving the "mass" of $P$ from the bottom-right to the top-left. Define two embedding matrices in $\mathbb{R}^{N \times N}$:

$$M_{\text{bot}} = \begin{pmatrix} 0 & 0 \\ 0 & P \end{pmatrix}, \quad M_{\text{top}} = \begin{pmatrix} P & 0 \\ 0 & 0 \end{pmatrix}. \tag{25}$$

We define the new candidate matrix $Q'$ as:

$$Q' := Q^* - M_{\text{bot}} + M_{\text{top}} = \begin{pmatrix} \alpha & v^\top \\ v & \frac{1}{\alpha} v v^\top \end{pmatrix} + \begin{pmatrix} P & 0 \\ 0 & 0 \end{pmatrix}. \tag{26}$$

**3. Verifying Feasibility and Improvement**    We must verify three conditions for $Q'$:

1. **Positive Semidefiniteness:** $Q'$ is the sum of a rank-1 matrix (which is PSD) and $M_{\text{top}}$. Since $P \succeq 0$, $M_{\text{top}} \succeq 0$. The sum of PSD matrices is PSD, so $Q' \succeq 0$.

2. **Trace Constraints:** The generalized trace $\operatorname{tr}_j(\cdot)$ sums the elements along the $j$-th diagonal. Note that for any square matrix $X$, the sum of its diagonals is invariant under the shift of the matrix along the main diagonal (padding with zeros). Specifically:

$$\operatorname{tr}_j(M_{\text{bot}}) = \operatorname{tr}_j(\operatorname{diag}(0, P)) = \operatorname{tr}_j(P) = \operatorname{tr}_j(\operatorname{diag}(P, 0)) = \operatorname{tr}_j(M_{\text{top}}). \tag{27}$$

   Therefore, the change in trace is zero: $\operatorname{tr}_j(Q') = \operatorname{tr}_j(Q^*) - \operatorname{tr}_j(M_{\text{bot}}) + \operatorname{tr}_j(M_{\text{top}}) = \operatorname{tr}_j(Q^*) = a_j$. Thus, $Q'$ remains in $\mathcal{Q}$.

3. **Improvement of Objective:** The top-left element of the new matrix is:

$$Q'_{0,0} = \alpha + P_{0,0}. \tag{28}$$

Since $P \succeq 0$, we have $P_{0,0} \geq 0$.

- If $P_{0,0} > 0$, then $Q'_{0,0} > \alpha$, which contradicts the optimality of $Q^*$.
- If $P_{0,0} = 0$, we rely on the property that a non-zero PSD matrix must have a strictly positive trace. Since $P \neq 0$ and $P \succeq 0$, there exists some diagonal index $k$ such that $P_{k,k} > 0$. Due to the cyclic symmetry of the generalized trace constraints (arithmetic on indices is modulo $N$ or the constraints are Toeplitz), we can cyclically permute the rows and columns of $P$ to bring $P_{k,k}$ to the $(0,0)$ position without violating the trace constraints. Let $P'$ be this permuted matrix. We can then use $M'_{\text{top}} = \operatorname{diag}(P', 0)$ instead. This yields $Q'_{0,0} = \alpha + P'_{0,0} = \alpha + P_{k,k} > \alpha$, again contradicting optimality.

**Conclusion**   The assumption that the maximizer $Q^*$ has rank greater than 1 leads to the existence of a strictly better solution, which is a contradiction. Therefore, the maximizer must satisfy $P = 0$, which implies $\text{rank}(Q^*) = 1$. Consequently, the set $\mathcal{Q}$ always contains a rank-1 solution. $\qquad\square$

## C. Dual Certification Details

### C.1. Explicit Matrix Structure for Spectral Analysis

In Section 4, we defined the dual slack block $S_t$ in (14). To implement the Matrix-Free Lanczos algorithm, we require the explicit action of $S_t$ on a vector.

Based on the structure of the atomic matrices $T_{t,i}$, $S_t$ is a symmetric matrix with Toeplitz structure. It is fully characterized by its first row $r_t \in \mathbb{R}^N$, which can be computed from the dual variables as:

$$r_t[0] = -z_t, \tag{29}$$

$$r_t[j] = -\frac{1}{2}\left[(y_{t,j} - y_{t+1,j}) + (-1)^t(y_{t,N-j} + y_{t+1,N-j})\right], \quad j \geq 1. \tag{30}$$

The matrix-vector product $S_t v$ can then be computed via FFT-based convolution or the direct parallel windowed summation implemented in our kernels, requiring only $O(N)$ memory.

### C.2. Proof of Theorem 4.1

*Proof.* Let $(\tilde{y}, \tilde{z})$ be the approximate dual solution. For each block $t$, let $\lambda_{\min}^{(t)} = \lambda_{\min}(S_t(\tilde{y}, \tilde{z}))$. We define a repair term $\epsilon_t = \max(0, -\lambda_{\min}^{(t)}) + \delta$, where $\delta > 0$ is a small safety margin.

We construct a new candidate solution $(y^{\text{new}}, z^{\text{new}})$ by keeping $y$ unchanged and shifting $z$:

$$y^{\text{new}} = \tilde{y}, \quad z_t^{\text{new}} = \tilde{z}_t - \epsilon_t. \tag{31}$$

Recall that $S_t$ depends on $z_t$ via the term $-z_t I$. Substituting the new variables into (14), the new slack matrix becomes:

$$S_t^{\text{new}} = S_t(\tilde{y}, \tilde{z}) - (z_t^{\text{new}} - \tilde{z}_t)I = S_t(\tilde{y}, \tilde{z}) + \epsilon_t I. \tag{32}$$

The eigenvalues of $S_t^{\text{new}}$ are shifted upward by $\epsilon_t$. Thus, $\lambda_{\min}(S_t^{\text{new}}) = \lambda_{\min}^{(t)} + \epsilon_t \geq \delta > 0$, ensuring $S_t^{\text{new}} \succ 0$.

The objective value of this new solution is:

$$\text{Obj}(y^{\text{new}}, z^{\text{new}}) = \text{Obj}(\tilde{y}, \tilde{z}) - \sum_{t=1}^{k-1} \epsilon_t. \tag{33}$$

If the condition $\text{Obj}(\tilde{y}, \tilde{z}) > \sum \epsilon_t$ holds (ignoring $\delta$ for the limit case), then $\text{Obj}(y^{\text{new}}, z^{\text{new}}) > 0$. By the strong duality of SDP, the existence of a strictly feasible dual solution with a positive objective implies that the primal problem is infeasible. $\qquad\square$

