# OpenReview forum: "Matrix-Free GPU Semidefinite Programming for Quantum Ordered Search at the k=6 Frontier"
_ICML.cc/2026/Conference — ICML 2026 regular_

### Official Review · Reviewer_k39S · 2026-03-12

**Soundness:** 3
**Presentation:** 2
**Significance:** 3
**Originality:** 3
**Overall Recommendation:** 4
**Confidence:** 1

**Summary:**

The paper addresses the Ordered Search Problem (OSP), a fundamental question in quantum information theory regarding the constant-factor speedup quantum algorithms offer over classical binary search. The core task is to find the largest admissible list size $N^*$ for a given number of quantum queries $k$, which is formulated as a Semidefinite Program (SDP). While previous research reached $k=5$ , the problem becomes intractable at $k=6$ because existing solvers require explicit construction of massive constraint matrices, hitting an insurmountable "memory wall". To overcome this, the authors introduce a domain-aware, matrix-free GPU SDP framework. By using custom CUDA kernels to evaluate highly structured constraints on-the-fly, they reduce memory complexity from $O(N^2)$ to $O(N)$.

**Compliance With Llm Reviewing Policy:**

Affirmed.

**Key Questions For Authors:**

While I am relatively new to this specific field, I would appreciate a clarification on the broader significance of the $k$ evolution ($k=5 \to 6 \to 7$). Specifically, what are the primary theoretical or practical milestones achieved by pushing the query depth to these new frontiers?

**Limitations:**

Computational Latency: The k=6 instance required approximately 16 days on a single GPU. This means that this method might require an unacceptable amount of computation for larger cases like k=7.

**Strengths And Weaknesses:**

While memory is saved, the computational cost is significant. Solving the $k=6$ instance required approximately 16 days of continuous execution on a H100 GPU.

---

> ### Author Rebuttal · Authors · 2026-03-30
>
> Thank you for your constructive review and for highlighting the core memory optimization achieved by our matrix-free framework. We appreciate the opportunity to clarify the broader context and significance of the problem, as well as the path forward for larger scale instances.
>
> **1. The Broader Significance of the $k$ Evolution ($k=5 \to 6 \to 7$)**
> This is an excellent question that strikes at the heart of the quantum query complexity field.
>
> *   **Closing the Theoretical Gap:** In classical computer science, exact binary search takes exactly $\log_2 N$ queries. A fundamental open question in quantum algorithms is determining the exact leading constant $c$ for the quantum query complexity $Q(N) \approx c \log_2 N$. While theoretical lower bounds dictate that $c \ge 0.221$ (Høyer & Neerbek, 2001), the constructive way to find a tighter upper bound is to design exact quantum algorithms for a fixed number of queries $k$, find the maximum list size $N^\star(k)$ they can search, and compute $c_k = k / \log_2 N^\star(k)$.
> *   **Direct Algorithm Design:** Beyond merely calculating theoretical bounds, the significance of solving this SDP lies in actual algorithm construction. Solving the problem for larger $k$ enables us to explicitly design better and more efficient quantum ordered search algorithms. Pushing the query depth to these new frontiers represents a concrete contribution to the broader quantum computing field.
> *   **Pushing the Frontier and New Results:** Increasing $k$ tightens this upper bound and directly yields better algorithms. Prior milestones achieved $c \approx 0.433$ for $k=4$ and $c \approx 0.390$ for $k=5$. Our initial submission broke the $k=6$ barrier to yield $c \approx 0.365$. Furthermore, we would like to share an exciting update. We extended our computations and successfully bracketed the phase transition to a tighter window. As shown in the newly uploaded figure (accessible at https://anonymous.4open.science/r/test_figure-E7A8/), we observe clear feasibility at $N=92,500$ and an infeasible plateau at $N=92,600$. This new result directly tightens our upper bound on the quantum query coefficient to $c \le 6/\log_2(92,500) \approx 0.3637$. We will add a brief summary of this broader context in the Introduction to make the paper more accessible.
>
> **2. Computational Latency and Extensibility to larger $k$ (e.g., $k=7$)**
> You are entirely correct that 16 days of continuous computation on an H100 GPU is substantial. It is important to distinguish between the memory wall and the compute wall. We will explicitly include this in a new "Limitations and Future Work" section based on the following analysis:
> *   **Storage Cost and the Memory Wall:** Prior to our work, the $\mathcal{O}(N^2)$ memory requirement created a hard memory wall that far exceeded the capacity of any single modern GPU. This made the $k=6$ instance physically impossible to run using standard solvers. In our matrix-free formulation, the primal factor $R$ has dimensions $(k-1)N \times 1$ (since we employ a rank-1 factorization), and the dual variables $y$ require $\mathcal{O}(kN)$ memory. Therefore, the total memory footprint scales linearly as $\mathcal{O}(kN)$, successfully removing this hard memory barrier.
> *   **Computational Cost and the Compute Wall:** The total number of floating-point operations per iteration is $\mathcal{O}(kN^2)$ due to the implicit evaluation of the dense constraint operators (diagonal sums and convolutions). Through our custom CUDA kernels, this $\mathcal{O}(N^2)$ workload is mapped to highly parallel reductions, massively accelerating the wall-clock time but still resulting in significant computational latency for very large instances.
> *   **Scaling beyond $k=6$:** To obtain a better upper bound on the query coefficient $c$ than our current result ($c \approx 0.365$), the next milestone at $k=7$ would require proving feasibility for at least $N \ge 600,000$ (since $7 / \log_2(600,000) \approx 0.365$). We estimate that the optimal list size $N^\star$ for $k=7$ will actually exceed $1,000,000$. At this massive scale, the $\mathcal{O}(kN)$ memory requirement will still fit comfortably on a single modern GPU. However, the $\mathcal{O}(kN^2)$ floating-point operations will become a significant computational bottleneck. Fortunately, as shown in Figures 1 and 3, our operator evaluation can be perfectly decoupled into $N$ independent parallel streams. Therefore, the method is highly amenable to distributed multi-GPU environments. Tackling $k \ge 7$ will naturally require scaling out to large clusters, such as utilizing a 128-GPU computing node, along with dedicated multi-GPU algorithmic designs to efficiently handle cross-device communications and load balancing.
>
> ---
>
> **References**
>
> [1] P. Høyer and J. Neerbek. "Bounds on quantum ordered searching." *arXiv preprint quant-ph/0009032*, 2001.

---

### Official Review · Reviewer_EG8d · 2026-03-13

**Soundness:** 3
**Presentation:** 4
**Significance:** 4
**Originality:** 4
**Overall Recommendation:** 4
**Confidence:** 3

**Summary:**

The paper studies the Quantum Ordered Search Problem (OSP) and aims to improve upper bounds on the constant factor in the quantum query complexity. Prior work showed that designing a $k$-query quantum algorithm for ordered search can be formulated as a semidefinite program (SDP) whose feasibility determines the largest list size $N^{*}$ that can be searched with $k$ queries. However, solving this SDP becomes computationally intractable for $k \geq 6$ due to the need to explicitly construct extremely large constraint matrices.

To overcome this bottleneck, the paper proposes a matrix-free GPU-based augmented Lagrangian method that evaluates SDP constraint operators on-the-fly using custom CUDA kernels, reducing memory complexity from quadratic to linear. This approach enables solving the previously intractable $k=6$ instance on a single GPU. Using this framework, the authors show that the SDP is feasible for $N=90,000$ and provably infeasible for $N=94000$, thereby bracketing the optimal list size $90000 \leq N^* < 94000$ improving the best known upper bound on the query coefficient. The work also introduces a matrix-free spectral certification method to rigorously verify infeasibility.

**Compliance With Llm Reviewing Policy:**

Affirmed.

**Final Justification:**

The rebuttal sufficiently addressed my concerns. It reinforced my prior assessment.

**Key Questions For Authors:**

The experiment for $k=6$ takes roughly 16 days on an H100 GPU. Do the authors expect the same approach to extend to the $k=7$ case?

**Limitations:**

The paper does not explicitly discuss limitations. Some limitations that can be discussed:

1. The main result relies heavily on large-scale numerical optimization, and although the infeasibility result is supported by a dual certificate, the feasibility result is also not shown in a formal proof.

2. The computational cost for $k=6$ is indeed substantial (approximately 16 days).

**Strengths And Weaknesses:**

$\textbf{Soundness}$

The reviewer is not a domain expert in quantum algorithms, but the paper appears technically sound and well motivated based on the provided explanations. The problem formulation is clearly presented, and the connection between quantum ordered search algorithms and semidefinite programming (SDP) is well grounded in prior literature. The proposed matrix-free augmented Lagrangian framework is carefully motivated and the key algorithmic components are described in a principled way. The experimental evaluation is also well designed: the solver is first validated on known cases with $k \leq 5$ before being applied to the previously intractable $k=6$ case. In addition, the authors attempt to provide rigorous certification of infeasibility through a dual certificate and spectral repair procedure.

There are, however, a few caveats: The main result heavily relies on large-scale optimization instead of rigorous proof. The infeasibility at $N=94000$ is also certified numerically. There is limited discussion on solver stability involved in the optimization process.

$\textbf{Presentation}$

The paper is generally well written and clearly structured. The overall narrative is easy to follow.

Minor: $Tr_{i-N}$ -> $Tr_{N-i}$ in Eq.(3)?

$\textbf{Significance}$

The paper addresses an important and well-studied problem in quantum algorithms, namely improving the constant factor in the query complexity of the quantum ordered search problem. Determining tighter bounds for this coefficient has been a longstanding challenge, and the ability to push the computational frontier from $k\leq 5$ to $k=6$ represents a meaningful advance in this line of research.

$\textbf{Novelty}$

The paper’s novelty primarily from a computational perspective. The main technical contribution is a matrix-free GPU framework for solving the semidefinite programs arising in the quantum ordered search problem, which avoids explicitly constructing the extremely large constraint matrices that limit existing solvers. While the underlying SDP formulation and the connection between ordered search and semidefinite programming are well established in prior work, the matrix-free solver design and its successful application to this large-scale instance provide a novel computational contribution that advances the state of the art.

---

> ### Author Rebuttal · Authors · 2026-03-30
>
> Thank you for finding our presentation, significance, and originality to be excellent. We appreciate your thoughtful evaluation. We address your points below and will incorporate these clarifications into the revision.
>
> **1. Extensibility to $k=7$ and Computational Cost (Limitation 2)**
> You correctly note the substantial 16-day runtime for $k=6$. To obtain a better upper bound on the query coefficient $c$ than our current result ($c \approx 0.365$), the next milestone at $k=7$ would require proving feasibility for at least $N \ge 600,000$ (since $7 / \log_2(600,000) \approx 0.365$). We estimate that the optimal list size $N^\star$ for $k=7$ will exceed $1,000,000$. Our matrix-free design keeps memory manageable at $\mathcal{O}(kN)$ (requiring only 5 to 10 GB for $k=7$), but the computational workload scales as $\mathcal{O}(kN^2)$. A single GPU will be too slow. Fortunately, as shown in Figures 1 and 3, our operator evaluation can be perfectly decoupled into $N$ independent parallel streams. Therefore, the method is highly amenable to distributed multi-GPU environments. Tackling $k \ge 7$ will naturally require scaling out to large clusters, such as utilizing a 128-GPU computing node, along with dedicated multi-GPU algorithmic designs to efficiently handle cross-device communications and load balancing. We will include a brief discussion of this asymptotic scaling and future roadmap in the revised Conclusion.
>
> **2. Rigorous Proofs vs. Numerical Evidence (Limitation 1 & Caveat 1)**
> We completely agree on distinguishing rigorous proofs from numerical evidence.
> *   **Infeasibility:** Our spectral certification provides a mathematically rigorous computer-assisted proof by fully bounding floating-point inaccuracies.
> *   **Feasibility:** Generally speaking, there is no way to construct a strict feasibility certification. In the literature of quantum query complexity via SDPs, prior works have traditionally relied on high-precision numerical results to establish the feasibility of a $(k, N)$ pair. For example, Childs et al. (2007) accepted the numerical output of standard interior-point solvers (like SeDuMi or SDPT3) as sufficient evidence. Even the recent state-of-the-art work for $k=5$ by Carolan et al. (2025), which elegantly eliminates equality constraints analytically via LP relaxation, ultimately relies on finding the roots of the derivatives of high-degree Chebyshev polynomials to verify the non-negativity constraints. Since finding roots of such high-degree polynomials is intrinsically a numerical process, their validation fundamentally relies on high-precision numerical evidence as well.
>
> Following this established practice, our matrix-free solver initially achieved a maximum constraint violation on the order of $10^{-10}$ for $N=90,000$ (as detailed in Table 2). Furthermore, we would like to share an exciting update: we extended our computations and successfully bracketed the phase transition to an even tighter window. As shown in the newly uploaded figure (accessible at https://anonymous.4open.science/r/test_figure-E7A8/), we observe clear feasibility at $N=92,500$ and an infeasible plateau at $N=92,600$. This new result tightens our upper bound on the quantum query coefficient to $c \le 6/\log_2(92,500) \approx 0.3637$. We consider this high-precision convergence to be highly compelling numerical evidence of feasibility. We will clarify that the feasibility result is supported by strong numerical evidence in line with prior literature.
>
> **3. Solver Stability (Caveat 3)**
> While the ALM with Burer-Monteiro factorization is non-convex, the OSP problem possesses a highly favorable structure. Our solver requires no manual hyperparameter tuning. As shown in Figure 5, it exhibits uniform and predictable convergence across all tested scales ($k=2,3,4,5$). For these verified cases, the solver stably identifies the optimal $N^\star$ by demonstrating a clear empirical phase transition between $N^\star$ and $N^\star+1$. It reliably drops to numerical zero for feasible instances and hits a rigid plateau for infeasible ones. We will discuss this stability and tuning-free nature in the revision.
>
> **4. Minor Typo: Eq.(3) $Tr_{i-N}$ vs $Tr_{N-i}$**
>
> Thank you for the careful reading. By our definition in Eq.(4), $Tr_j(X)$ uses a negative index $j < 0$ for sub-diagonals. Since $i \in$ {1, ..., N-1}, $i-N$ is negative. Thus, $Tr_{i-N}(X)$ correctly refers to the $(N-i)$-th sub-diagonal (e.g., if $N=10$ and $i=1$, then $i-N = -9$, the 9th sub-diagonal). We will add a clarifying footnote to Eq.(3) to prevent confusion.
>
> ---
>
> **References**
>
> [1] A. M. Childs, A. J. Landahl, and P. A. Parrilo. "Quantum algorithms for the ordered search problem via semidefinite programming." *Phys. Rev. A*, 75:032335, 2007.
>
> [2] J. Carolan, A. M. Childs, M. Kovacs-Deak, and L. Schaeffer. "Translation-invariant quantum algorithms for ordered search are optimal." *arXiv preprint arXiv:2503.21090*, 2025.

---

> > ### Author Rebuttal · Reviewer_EG8d · 2026-04-01
> >
> > Thank you for the thoughtful response. My concerns have been fully addressed. I will maintain my score of 4.

---

### Official Review · Reviewer_pTiV · 2026-03-18

**Soundness:** 3
**Presentation:** 3
**Significance:** 2
**Originality:** 2
**Overall Recommendation:** 4
**Confidence:** 2

**Summary:**

This paper studies large-scale semidefinite programming (SDP) with a focus on overcoming the memory bottleneck that arises in GPU-based solvers. The key observation is that existing large-scale SDP approaches still require explicit construction or storage of the constraint operator, which becomes infeasible at scale. To address this, the authors propose a matrix-free augmented Lagrangian framework implemented on GPUs. Instead of materializing the constraint matrix, the method evaluates linear operators on the fly via custom CUDA kernels, thereby shifting the computational bottleneck from memory bandwidth to arithmetic throughput. The approach is applied to a large-scale SDP arising from the quantum ordered search problem (OSP). The proposed method enables solving instances at a scale (e.g., $k=6$) that were previously intractable due to memory limitations, thereby improving bounds on the associated quantum query complexity coefficient.

**Compliance With Llm Reviewing Policy:**

Affirmed.

**Final Justification:**

The authors addressed my main concerns. I maintain my positive score.

**Key Questions For Authors:**

See the weaknesses section.

**Strengths And Weaknesses:**

Strengths

1. The paper identifies a fundamental limitation of current large-scale SDP solvers: the need to explicitly store constraint matrices, which becomes prohibitive on GPUs. The matrix-free formulation is a natural and practically important direction.

2. The idea of replacing memory-bound operations with compute-bound operator evaluations is well aligned with modern GPU hardware characteristics. The use of custom kernels to evaluate structured constraints on-the-fly is a technically sound systems contribution.

3. The application to quantum ordered search is nontrivial and provides a compelling testbed where standard methods fail due to memory constraints. The ability to push the computational frontier (e.g., resolving the $k=6$ regime) is a notable empirical achievement.


Weaknesses:

1. The core optimization method is an augmented Lagrangian approach, which is standard in large-scale convex optimization. The main novelty lies in the implementation (matrix-free operator evaluation) rather than new algorithmic ideas.

2. The paper does not provide convergence or complexity guarantees for the proposed matrix-free ALM in this setting. Given that the method operates in a highly nonstandard regime, it would be important to understand its theoretical properties or at least provide justification.

3. The paper discusses prior GPU-based methods, but the experimental section does not provide a thorough empirical comparison.

---

> ### Author Rebuttal · Authors · 2026-03-30
>
> Thank you for your constructive review and for recognizing that our matrix-free formulation effectively overcomes fundamental memory limitations, perfectly aligning with modern GPU architectures to push the computational frontier of the quantum ordered search problem to $k=6$. We address your concerns below.
>
> **W1: Algorithmic novelty vs. implementation.**
> We agree with the reviewer that our algorithm is indeed built upon the standard Augmented Lagrangian Method (ALM) framework. The primary motivation of our work is to address the severe memory bottleneck encountered when solving the quantum ordered search problem (OSP) at $k=6$. To overcome this, we exploit the highly specific algebraic structure of the OSP semidefinite program to propose the matrix-free operator evaluation. By theoretically decomposing the constraints (Eqs. 10 to 13), we enable their implicit evaluation via $\mathcal{O}(N)$ streaming. This structure-aware design is exactly what allows us to scale the standard ALM framework to handle previously intractable problem sizes.
>
> **W2: Convergence and complexity guarantees.**
> This is an excellent point that we will clarify in the revision. Our matrix-free operator evaluation is **algebraically exact** (up to standard floating-point precision). We do not introduce any mathematical approximations to the linear operators $\mathcal{A}$ and $\mathcal{A}^*$. Therefore, the sequence of iterates generated by our matrix-free ALM is mathematically identical to the sequence that a standard matrix-based ALM would produce if infinite memory were available.
> Consequently, our method directly inherits all the existing theoretical convergence guarantees of the Burer-Monteiro based Augmented Lagrangian Method (e.g., convergence to stationary points), which have been established in the original literature [1] and extensively studied in recent works such as [2]. We will add a theoretical remark in Section 3.1 explicitly stating that the exactness of our matrix-free operators ensures the preservation of the base ALM's convergence properties.
>
> **W3: Empirical comparison with prior GPU-based methods.**
> We appreciate the suggestion. To address this, we would like to first clarify that recent GPU-based state-of-the-art solvers (cuLoRADS [3], ALORA [2], and cuHALLaR [4]) are all fundamentally based on the same Burer-Monteiro factorization combined with the Augmented Lagrangian Method (BM+ALM) framework. They differ primarily in their specific strategies for solving the unconstrained subproblems.
> Furthermore, the primary goal of our work is to determine the exact boundary of feasible and infeasible list sizes $N$ for the $k=6$ instance. At this target scale ($N \approx 10^5$), the original versions of all these general-purpose solvers require explicitly instantiating the constraint matrix $\mathcal{A}$, which inevitably leads to an immediate Out-of-Memory (OOM) error on any single modern GPU. To use ALORA or cuHALLaR for this specific problem, one would need to modify their source codes to integrate our custom matrix-free CUDA operators. However, we were unable to obtain the source codes for ALORA and cuHALLaR, making this modification and subsequent comparison practically impossible.
>
> ---
>
> **References**
>
> [1] S. Burer and R. D. Monteiro, "A nonlinear programming algorithm for solving semidefinite programs via low-rank factorization," *Math. Program.* 95(2):329-357, 2003.
>
> [2] L. Ding, H. Lu, and J. Yang, "New understandings and computation on augmented lagrangian methods for low-rank semidefinite programming," *arXiv preprint arXiv:2505.15775*, 2025.
>
> [3] Q. Han, Z. Lin, H. Liu, C. Chen, Q. Deng, D. Ge, and Y. Ye, "Accelerating low-rank factorization-based semidefinite programming algorithms on gpu," *arXiv preprint arXiv:2407.15049*, 2024.
>
> [4] J. M. Aguirre, D. Cifuentes, V. Guigues, R. D. Monteiro, V. H. Nascimento, and A. Sujanani, "cuhallar: A gpu accelerated low-rank augmented lagrangian method for large-scale semidefinite programming," *arXiv preprint arXiv:2505.13719*, 2025.

---

> > ### Author Rebuttal · Reviewer_pTiV · 2026-04-02
> >
> > Thanks for your response. I maintain my positive score.

---

> > > ### Author Response · Authors · 2026-04-05
> > >
> > > Thank you for your thoughtful feedback and for confirming that your concerns have been fully addressed. We truly appreciate your time and engagement in the review process.
> > >
> > > BTW, we have some new numerical results (accessible at https://anonymous.4open.science/r/test_figure-E7A8/), which suggest that $92,500 \le N^\star \le 92,600$ and $c \le 6 / (\log_2 92,500) \approx 0.364$. If you feel that the clarifications and the addition result strengthen the contribution, we would be grateful if you could consider raising the score.
> > >
> > > Thank you again for your support.

---

### Official Review · Reviewer_zg4i · 2026-03-24

**Soundness:** 3
**Presentation:** 3
**Significance:** 2
**Originality:** 3
**Overall Recommendation:** 4
**Confidence:** 4

**Summary:**

This paper investigates the fundamental query complexities of the quantum search problem, where the goal is to locate the index of a given target in a sorted array, using a quantum comparison oracle that can be queried in superposition. While quantum algorithms do not change the asymptotic log_2 n query complexity of the classical binary search, they can improve the leading constant. The authors seek to sharpen the best known upper bound on this constant by computing the largest array size that can be solved exactly with six quantum queries. Via a recursive construction, this finite k result yields algorithms for arbitrarily large arrays. They cast the problem as a semidefinite program and develop a matrix-free GPU implementation that overcomes the prohibitive memory requirements of prior methods, enabling computation in the challenging k=6 regime.

**Compliance With Llm Reviewing Policy:**

Affirmed.

**Final Justification:**

We thank the authors for their detailed clarifications, particularly regarding the rigor of the claims. As, this point is not yet reflected in the current manuscript. I would be supportive of the paper provided that the authors revise the presentation to clearly distinguish the feasibility component (i.e., the lower bound on N^*) as a numerical observation or conjecture, rather than a formally established result.

**Key Questions For Authors:**

The presentation could be improved by making several technical details more explicit so that the paper is more self-contained.

1. Please clarify the precise algorithmic setting. Is the quantum ordered search model here restricted to exact algorithms with zero error, or is any form of randomness and bounded error allowed?
2. For the feasible case of N=90,000, please explain more rigorously how the numerical observation is converted into a valid feasibility bound. The paper gives a clearer account of how this is achieve for the infeasibility bound, but the analogous justification for the feasibility claim is much less explicit.
3. What are the asymptotic storage and computational costs of the proposed method as functions of k? This would help readers assess how far the approach may scale beyond the k=6 case.

**Limitations:**

This paper primarily focuses on a theoretical question in quantum query complexity. In that sense, the discussion of social impact is appropriate for the scope of the work.

**Strengths And Weaknesses:**

Strength:

This paper studies a fundamental problem in quantum computation, and the authors make concrete progress toward tightening the best known leading constant in the query complexity of the quantum search problem.


Weakness

One limitation of the work is that, although it addresses a fundamental problem, the query complexity can only be improved by a constant factor. Moreover, even within this constant-factor regime and for the k=6 special case, the presented result is not an exact characterization, but rather a certified range. Besides, as the paper centers on a numerical search procedure and its implementation, it would be helpful for the authors to explain more explicitly how the numerical results translate into rigorous guarantees. The detailed suggestions are in the Key Questions section.

---

> ### Author Rebuttal · Authors · 2026-03-30
>
> Thank you for your constructive feedback and for recognizing the fundamental nature of the problem we study. We address your specific questions below and will update the manuscript to make these technical details more explicit.
>
> **Q1: Precise algorithmic setting (Exact vs. Bounded Error)**
> Our setup and the resulting SDP formulation strictly follow the framework established by Childs et al. (2007), which focuses specifically on the **exact** quantum ordered search algorithm (i.e., zero error). We will explicitly clarify in Section 2.1 that our model and results apply strictly to exact algorithms.
>
> **Q2: Rigorous guarantee for the feasible case ($N=90,000$)**
> You are correct to distinguish between the formally certified infeasibility bound and the feasibility claim.
> *   **Infeasibility:** We formally verify this via a computer-assisted proof using spectral shifting, which constructs a rigorous dual certificate despite floating-point arithmetic.
> *   **Feasibility:** Generally speaking, there is no way to construct a strict feasibility certification. In the literature of quantum query complexity via SDPs, prior works have traditionally relied on high-precision numerical results to establish the feasibility of a $(k, N)$ pair. For example, Childs et al. (2007) accepted the numerical output of standard interior-point solvers (like SeDuMi or SDPT3) as sufficient evidence. Even the recent state-of-the-art work for $k=5$ by Carolan et al. (2025), which elegantly eliminates equality constraints analytically via LP relaxation, ultimately relies on finding the roots of the derivatives of high-degree Chebyshev polynomials to verify the non-negativity constraints. Since finding roots of such high-degree polynomials is intrinsically a numerical process, their validation fundamentally relies on high-precision numerical evidence as well.
>
> Following this established practice, our matrix-free solver initially achieved a maximum constraint violation on the order of $10^{-10}$ for $N=90,000$ (as detailed in Table 2). Furthermore, we extended our computations and successfully bracketed the phase transition to an even tighter window. As shown in the newly uploaded figure (accessible at https://anonymous.4open.science/r/test_figure-E7A8/), we observe clear feasibility at $N=92,500$ and an infeasible plateau at $N=92,600$. This new result tightens our upper bound on the quantum query coefficient to $c \le 6/\log_2(92,500) \approx 0.3637$. We consider this high-precision convergence to be highly compelling numerical evidence of feasibility. We will clarify that the feasibility result is supported by strong numerical evidence in line with prior literature.
>
> **Q3: Asymptotic storage and computational costs as functions of $k$**
> This is crucial for understanding the scalability limits of our framework.
> *   **Storage Cost:** In our matrix-free formulation, the primal factor $R$ has dimensions $(k-1)N \times 1$ (since we employ a rank-1 factorization, $r=1$), and the dual variables $y$ require $\mathcal{O}(kN)$ memory. Therefore, the total memory footprint scales linearly as $\mathcal{O}(kN)$.
> *   **Computational Cost:** The total number of floating-point operations per iteration is $\mathcal{O}(kN^2)$ due to the implicit evaluation of the dense constraint operators (diagonal sums and convolutions). Through our custom CUDA kernels, this $\mathcal{O}(N^2)$ workload is mapped to highly parallel reductions, massively accelerating the wall-clock time.
> *   **Scaling beyond $k=6$:** To obtain a better upper bound on the query coefficient $c$ than our current result ($c \approx 0.365$), the next milestone at $k=7$ would require proving feasibility for at least $N \ge 600,000$ (since $7 / \log_2(600,000) \approx 0.365$). We estimate that the optimal list size $N^\star$ for $k=7$ will exceed $1,000,000$. At this massive scale, the $\mathcal{O}(kN)$ memory requirement will still fit comfortably on a single modern GPU. However, the $\mathcal{O}(kN^2)$ floating-point operations will become a significant computational bottleneck. Fortunately, as shown in Figures 1 and 3, our operator evaluation can be perfectly decoupled into $N$ independent parallel streams. Therefore, the method is highly amenable to distributed multi-GPU environments. Tackling $k \ge 7$ will naturally require scaling out to large clusters, such as utilizing a 128-GPU computing node, along with dedicated multi-GPU algorithmic designs to efficiently handle cross-device communications and load balancing. We will include a brief discussion of this asymptotic scaling and future roadmap in the revised Conclusion.
>
> ---
>
> **References**
>
> [1] A. M. Childs, A. J. Landahl, and P. A. Parrilo. "Quantum algorithms for the ordered search problem via semidefinite programming." *Phys. Rev. A*, 75:032335, 2007.
>
> [2] J. Carolan, A. M. Childs, M. Kovacs-Deak, and L. Schaeffer. "Translation-invariant quantum algorithms for ordered search are optimal." *arXiv preprint arXiv:2503.21090*, 2025.

---

> > ### Author Rebuttal · Reviewer_zg4i · 2026-04-04
> >
> > Thank you for the detailed response. My question regarding the rigorous guarantee of feasibility remains. For any statement presented as a formal theoretical claim, it would need to be supported by the explicit steps that justifies why the achieved numerical accuracy implies theoretical correctness. This step is essential and cannot be replaced by convention alone. This justification may be straightforward, but it is important to make it explicit.

---

> > > ### Author Response · Authors · 2026-04-07
> > >
> > > We thank the reviewer for raising this important point. We fully agree that a small numerical residual alone does not constitute a formal mathematical proof of exact feasibility. In the revised manuscript, we will strictly distinguish our rigorously certified **infeasibility** results from our **feasibility** bounds, which will be explicitly framed as strong empirical evidence rather than formal theorems.
> > >
> > > Below, we detail our clarification of claims, explain why the observed numerical signature remains highly convincing, and discuss the theoretical pathway needed to rigorously prove feasibility.
> > >
> > > ### 1. Clarification of Claims
> > > Our rigorous theoretical guarantee in the current paper is one-sided: it applies strictly to the **infeasibility** bound. Specifically, for the $N = 94,000$ cases, we provide an explicit dual certificate and apply a spectral-shifting repair argument. This yields a mathematically rigorous proof of infeasibility that is robust to floating-point arithmetic.
> > >
> > > By contrast, for $N = 90,000$, our solver achieves extremely small residuals (relative infeasibility $7.49 \times 10^{-10}$ and maximum violation $1.99 \times 10^{-10}$), but lacks an a posteriori certification argument. We agree that without such an argument, this does not formally guarantee the existence of an exact feasible SDP point.
> > >
> > > Therefore, we will revise the paper to explicitly state that our formally certified theoretical claim is the upper bound $N^* < 94,000$. The lower bound $N \ge 90,000$ will be rephrased and explicitly presented as being supported by **strong numerical evidence**, rather than as a formally proven theorem.
> > >
> > > ### 2. Why the Numerical Evidence is Highly Convincing
> > > While empirical, we interpret the numerical accuracy at $N=90,000$ as a highly reliable indicator of true feasibility due to the consistent, signature phase transitions observed across our solver for all known cases up to $k=5$.
> > >
> > > As illustrated in Figure 5, the geometry of the Augmented Lagrangian Method creates a stark dichotomy:
> > > *   For every known, exact optimal list size $N^*$, the primal infeasibility smoothly converges to numerical zero (well below $10^{-8}$).
> > > *   In sharp contrast, evaluating the immediately infeasible instance $N^*+1$ consistently results in an early stagnation at a strictly positive "infeasibility plateau."
> > >
> > > The convergence trajectory for $k=6, N=90,000$ cleanly avoids this plateau and reaches the $10^{-10}$ accuracy regime, perfectly matching the behavioral signature of all verified feasible instances. We will include this specific contextual justification in the revised text to explain why the numerical result strongly points to exact feasibility.
> > >
> > > ### 3. The Theoretical Pathway to Rigorous Feasibility
> > > For completeness, we would also like to discuss exactly what would be required to translate our high-accuracy numerical solution ($\hat{X} \succeq 0, \\|\mathcal{A}\hat{X} - b\\| \le \varepsilon$) into a rigorous proof of exact feasibility, and why we leave this for future work.
> > >
> > > The theoretical bridge relies on the Approximate Farkas Lemma for conic systems [1, 2]. Consider our primal feasibility problem:
> > > $$
> > > \min_X \quad 0 \quad \text{subject to} \quad \mathcal{A} X = b, \quad X \succeq 0
> > > $$
> > > Its strong alternative (the dual certificate of infeasibility) is:
> > > $$
> > > \max_y \quad 0 \quad \text{subject to} \quad \mathcal{A}^*y \preceq 0, \quad b^\top y = 1
> > > $$
> > > Let $\gamma := \min \\{\\|v\\| \mid \mathcal{A}X - b = v , X \succeq 0 \\}$ be the distance to primal feasibility. According to the Approximate Farkas Lemma (e.g., Lemma 2.2 of [1]), if the dual alternative is feasible, the minimum-norm dual certificate $y^\star_m$ satisfies:
> > > $$
> > > \\|y^\star_m\\| = \frac{1}{\gamma}
> > > $$
> > > Given our numerical solution $\hat{X} \succeq 0$, we know that $\gamma \leq \\|\mathcal{A}\hat{X} - b\\| = \varepsilon$. Therefore, if the primal were truly infeasible, there *must* exist a dual certificate $y^\star_m$ such that:
> > > $$
> > > \\|y^\star_m\\| = \frac{1}{\gamma} \geq \frac{1}{\varepsilon}
> > > $$
> > > **To formulate a rigorous proof:** If one could analyze the algebraic structure of the operator $\mathcal{A}$ to establish an absolute theoretical upper bound $C$ on the norm of *any* valid dual certificate (i.e., proving that if a dual certificate exists, it must satisfy $\\|y^\star_m\\| \leq C$), then achieving a numerical residual $\varepsilon < \frac{1}{C}$ would mathematically prove that no such dual certificate can exist. By the theorem of strong alternatives, this would strictly certify exact primal feasibility.
> > >
> > > ---
> > >
> > > **References:**
> > >
> > > [1] Todd, M. J., & Ye, Y. (1998). Approximate Farkas lemmas and stopping rules for iterative infeasible-point algorithms for linear programming. *Mathematical Programming*, *81*(1), 1-21.
> > >
> > > [2] Pólik, I., & Terlaky, T. (2009). New stopping criteria for detecting infeasibility in conic optimization. *Optimization Letters*, *3*(2), 187-198.

---

### Decision · Program_Chairs · 2026-04-30

**Decision:**

Accept (regular)

**Comment:**

This paper presents a significant computational advancement in the study of the Quantum Ordered Search Problem (OSP) by overcoming the "memory wall" that previously limited research to $k=5$. The authors introduce a matrix-free GPU semidefinite programming (SDP) framework that evaluates highly structured constraints on-the-fly using custom CUDA kernels. This innovation reduces memory complexity from quadratic to linear, shifting the primary bottleneck from memory capacity to computational throughput.

The technical and scientific contributions are well-supported:

Frontier Expansion: The framework allows the calculation of the $k=6$ case on a single modern GPU, which was previously intractable due to the prohibitively large constraint matrices required by existing solvers.

Improved Bounds: The authors tightly bracket the optimal list size $N^{*}$ for $k=6$ between 90,000 and 94,000. This improves the best known upper bound on the quantum query coefficient from 0.390 to 0.365. Supplemental results provided during the rebuttal further refined this bound to approximately 0.3637.

Rigorous Certification: The paper introduces a matrix-free spectral certification method to provide mathematically rigorous dual infeasibility certificates, ensuring the results are robust against floating-point inaccuracies.

Response to Reviewer ConcernsThe reviewers initially raised valid concerns regarding the rigor of the feasibility claims and the scalability of the approach. The authors successfully addressed these points through the rebuttal process:

Rigor of Feasibility: In response to Reviewer zg4i, the authors clarified the distinction between the formally certified infeasibility bound ($N < 94,000$) and the feasibility claim ($N = 90,000$), which is supported by strong numerical evidence consistent with established literature. The reviewers expressed satisfaction with this clarification.

Scalability: While the computational cost for $k=6$ is high (approximately 16 days on an H100 GPU), the authors provided a clear roadmap for scaling to $k=7$ using distributed multi-GPU environments, noting that the linear memory scaling remains manageable.

Algorithmic Novelty: While the core optimization uses a standard Augmented Lagrangian Method, the reviewers agreed that the matrix-free implementation and its successful application to a fundamental quantum computing problem represent a novel and valuable contribution.

Overall, the submission is technically solid and provides a meaningful advance in both large-scale optimization and quantum query complexity. The consensus among reviewers is positive, with multiple "Weak Accept" ratings and high scores for significance and originality.